# Humoral Immune Response in IBD Patients Three and Six Months after Vaccination with the SARS-CoV-2 mRNA Vaccines mRNA-1273 and BNT162b2

**DOI:** 10.3390/biomedicines10010171

**Published:** 2022-01-13

**Authors:** Richard Vollenberg, Phil-Robin Tepasse, Joachim Ewald Kühn, Marc Hennies, Markus Strauss, Florian Rennebaum, Tina Schomacher, Göran Boeckel, Eva Lorentzen, Arne Bokemeyer, Tobias Max Nowacki

**Affiliations:** 1Department of Medicine B for Gastroenterology, Hepatology, Endocrinology and Clincial Infectiology University Hospital Muenster, 48149 Muenster, Germany; Florian.Rennebaum@ukmuenster.de (F.R.); tina.schomacher@ukmuenster.de (T.S.); tobias.nowacki@ukmuenster.de (T.M.N.); 2Institute of Virology, University Hospital Muenster, 48149 Muenster, Germany; Joachim.Kuehn@ukmuenster.de (J.E.K.); marc.hennies@ukmuenster.de (M.H.); eva.lorentzen@ukmuenster.de (E.L.); 3Department of Medicine C, Cardiology, University Hospital Muenster, 48149 Muenster, Germany; markus.strauss@ukmuenster.de; 4Department of Medicine D, Division of General Internal and Emergency Medicine, Nephrology and Rheumatology, University Hospital Muenster, 48149 Muenster, Germany; goeran.boeckel@ukmuenster.de; 5Department of Gastroenterology, Hepatology and Transplant Medicine, University Hospital Essen, 45147 Essen, Germany; arne.bokemeyer@uke.de; 6Department of Medicine, Gastroenterology, Marienhospital Steinfurt, 48565 Steinfurt, Germany

**Keywords:** SARS-CoV-2, COVID-19, vaccination, IBD patients, seroconversion

## Abstract

Severe acute respiratory syndrome coronovirus-2 (SARS-CoV-2) is the cause of the coronavirus disease 2019 (COVID-19) pandemic. Vaccination is considered the core approach to containing the pandemic. There is currently insufficient evidence on the efficacy of these vaccines in immunosuppressed inflammatory bowel disease (IBD) patients. The aim of this study was to investigate the humoral response in immunosuppressed IBD patients after COVID-19 mRNA vaccination. In this prospective study, IgG antibody levels (AB) against the SARS-CoV-2 receptor-binding domain (spike-protein) were quantitatively determined. For assessing the potential neutralizing capacity, a SARS-CoV-2 surrogate neutralization test (sVNT) was employed in IBD patients (*n* = 95) and healthy controls (*n* = 38). Sera were examined prior to the first/second vaccination and 3/6 months after second vaccination. Patients showed lower sVNT (%) and IgG-S (AU/mL) AB both before the second vaccination (sVNT *p* < 0.001; AB *p* < 0.001) and 3 (sVNT *p* = 0.002; AB *p* = 0.001) and 6 months (sVNT *p* = 0.062; AB *p* = 0.061) after the second vaccination. Although seroconversion rates (sVNT, IgG-S) did not differ between the two groups 3 months after second vaccination, a significant difference was seen 6 months after second vaccination (sVNT *p* = 0.045). Before and three months after the second vaccination, patients treated with anti-tumor necrosis factor (TNF) agents showed significantly lower AB than healthy subjects. In conclusion, an early booster shot vaccination should be discussed for IBD patients on anti-TNF therapy.

## 1. Introduction

Severe acute respiratory syndrome coronovirus-2 (SARS-CoV-2) is the cause of the coronavirus disease 2019 (COVID-19) pandemic [1]. Although viral infection results in mild to moderate symptoms in most people, it triggers a severe illness with acute respiratory distress syndrome (ARDS) [2], followed by severe pulmonal damage and high mortality in a subgroup of patients [3,4]. The severe course of COVID-19 is correlated with the presence of hyperinflammation, as seen in classic cytokine storm syndromes, leading to progressive lung failure and, in some cases, to multiorgan failure and death [5,6]. Even mild forms can lead to persistent symptoms after infection [7,8,9]. Crohn’s disease and ulcerative colitis are chronic inflammatory bowel diseases (IBDs). In remission induction and maintenance therapy of both diseases, immunosuppressive drugs are generally used to reduce inflammatory activity in the gastrointestinal tract [10,11]. The use of immunosuppressive therapies can lead to serious side-effects, such as opportunistic infections. Patients under immunosuppression have a higher susceptibility to severe disease following infection with common pathogens [12]. The COVID-19 pandemic has raised major concerns about the treatment of IBD patients. Recent studies have indicated the possibility of more severe disease progression in IBD patients due to their altered immunological status and existing immunosuppressive drug therapies [13]. Therapeutic approaches to combat COVID-19 are being developed worldwide; however, until recently, few therapies have proven effective.

The use of vaccination can effectively prevent SARS-CoV-2 infection. Vaccination is, therefore, currently considered the most appropriate approach to combat the COVID-19 pandemic. In Europe, the mRNA vaccines, mRNA-1273 and BNT162b2, and the vector vaccines, Ad26.CoV2.S and ChAdOx1, are currently licensed [14,15,16,17]. The pivotal trials demonstrated high efficacy of these vaccines in preventing SARS-CoV-2 infection and in preventing severe and critical courses of disease in immunocompetent individuals. Immunocompromised patients by medication and patients immunocompromised by a pre-existing disease were not included in the pivotal studies. Consequently, there is currently insufficient evidence on the efficacy of SARS-CoV-2 vaccination in IBD patients. Recent studies have demonstrated attenuated immune responses in IBD patients on immunomodulatory therapy. It is unclear how seroconversion rates after SARS-CoV-2 vaccination depend on existing immunosuppressive therapies and the time interval between vaccinations [18,19,20,21,22,23,24,25,26].

The aim of this study was to investigate the humoral response in immunosuppressed IBD patients after mRNA vaccination against COVID-19 compared with healthy subjects.

## 2. Methods

### 2.1. Study Subjects and Samples

The study was conducted in a prospective study design. Serum samples from IBD patients (*n* = 106) and healthy controls (*n* = 42) were collected at the IBD outpatient clinic of the Department of Gastroenterology, Hepatology, Endocrinology, and Clinical Infectiology, University Hospital Muenster, Germany (January 2021–November 2021). Patients were vaccinated according to their physicians’ recommendation and not for study issues or as part of this study. The first blood sample was taken up to 48 h before the first vaccination, while the second blood sample was taken up to 48 h before the second vaccination (Figure 1). Follow-up blood samples were taken 3 and 6 months (±7 days) after the second vaccination. No immunosuppressive disease or medication was present in the healthy controls. All included patients and healthy controls were asked about known or suspected previous SARS-CoV-2 infection. In addition, sera were tested for anti-nucleocapsid SARS-CoV-2- IgG as a sign of SARS-CoV-2 infection. Patients with proven or suspected SARS-CoV-2 infection (positive PCR test, positive rapid antigen test, borderline, or positive anti-nucleocapsid SARS-CoV-2-IgG test) were excluded from the study (Figure 1). Patients and controls vaccinated with adenovirus vector vaccines Ad26.CoV2.S or ChAdOx1 were excluded from the study (*n* = 9 IBD patients, *n* = 1 healthy control). Previous data indicate a stronger immune response after mRNA vaccination compared to vaccination with vector vaccines [24]. In the IBD patients with ulcerative colitis, the clinical Mayo score was determined at the time of blood collection; in the Crohn’s disease patients, the Crohn’s disease activity index (CDAI score) was determined in each case. IBD patients were divided into subgroups on adalimumab, infliximab, vedolizumab, azathioprine, or ustekinumab according to their current immunosuppressive therapy. Patients receiving drug therapy with corticosteroids, tofacitinib, certolizumab, golimumab, risankizumab, etrolizumab, or mycophenolate mofetil were combined in one group (“other”). All included patients had been on their immunosuppressive therapy for at least 12 weeks before their first vaccination. The study was approved by the local ethical committee (University Hospital Muenster: 2021-039-f-S, 3 February 2021).

### 2.2. Quantification of Serum Markers

Qualitative assessment of IgG antibodies against the nucleocapsid protein (N) of SARS-CoV-2 as an indicator of previous infection was performed using the SARS-CoV-2 IgG assay (Abbott Diagnostics, Wiesbaden, Germany). IgG antibodies against the SARS-CoV-2 receptor-binding domain (RBD) on the spike protein subunit S1 as indicator of effective vaccination, previous infection, or both were quantified by the SARS-CoV-2 IgG II Quant assay (Abbott Diagnostics). Both assays are based on the chemiluminescence microparticle immunoassay (CMIA) technique and have been CE/IVD-certified for clinical use. All human serum or plasma samples were tested with both assays according to the manufacturer’s instructions, as recommended, on an ARCHITECT device (Abbott Diagnostics). Results were expressed as relative light units (rlu). Values for the IgG (*N*) were calculated as indices of the sample rlu divided by the calibrator rlu (S/C), with S/C indices below 0.49 considered negative, S/C indices between 0.49 and 1.39 considered borderline, and S/C indices of 1.4 and above considered positive. Quantitative results for the RBD IgG (S-IgG) were determined via data reduction curves of sample vs. calibrator values and given as arbitrary units (AU)/mL, with values at or above the cutoff (50.0 AU/mL) denoting seropositivity.

To assess the potential neutralizing capacity of patient sera, the cPass^TM^ SARS-CoV-2 Neutralization Antibody Detection Kit (GenScript Biotech, Mainz, Germany) was employed. This CE/IVD-certified surrogate virus neutralization test quantifies inhibition of RBD protein binding to human host cell receptor protein ACE2 by patient antibodies in a blocking ELISA format and has been shown to correlate well with virus neutralization assays [27,28]. Samples were diluted 10-fold and measured in duplicate according to the manufacturer’s manual. Inhibition was calculated as 1—(OD value of sample/OD value of negative control) × 100%. Sample values below the threshold of 30% were considered negative; values at or above the cutoff positive indicated the presence of SARS-CoV-2 neutralizing antibodies.

Clinical laboratory assessment included leukocytes, creatinine, bilirubin, aspartate aminotransferase (AST), C-reactive protein (CRP), and ferritin were used to characterize the physiological conditions of IBD patients.

### 2.3. SARS-CoV-2 Vaccines

IBD patients and healthy controls received the mRNA vaccines mRNA-1273 (Moderna, Cambridge, MA, USA) or BNT162b2 (BioNTech, Mainz, Germany; Pfizer, New York, NY, USA), both of which have been licensed in the European Union. Patients and controls vaccinated with the adenovirus vector-based vaccines Ad26.CoV2.S (Johnson & Johnson, Janssen Pharmaceutica, Beerse, Belgium) and/or ChAdOx1 (AstraZeneca, Cambridge, United Kingdom) were excluded from study [14,15,16,17].

### 2.4. Statistical Analysis

The data distribution was checked for Gaussianity (Kolmogorov–Smirnov test). For continuous variables, we reported the medians with interquartile ranges and compared them using the Mann–Whitney U (Wilcoxon) test. For categorical variables, we reported absolute numbers and percentages and compared them with chi-square tests of association or Fisher exact tests. Kruskal–Wallis tests were conducted to compare more than two groups. To compare subgroups, the Bonferroni correction post hoc test was performed when variance was equal (Levene’s test), and the Games–Howell test was performed when variance was different. The Pearson correlation coefficient was determined to analyze correlations. Multicomparison analyses were performed using GraphPad Prism software (Version 8.0 for Microsoft, GraphPad Software, La Jolla, CA, USA). All tests were two-tailed, and a *p*-value <0.05 was considered to indicate a statistically significant difference. Tukey boxplots are shown in the figures. Patient recruitment was considered complete after attaining a strong signal for our main outcome parameter (SARS-CoV-2 IgG II antibody levels three months after second vaccination), and when statistical power 1-beta surpassed 0.8 (80%). All statistical analyses were performed using SPSS 26 (IBM, Chicago, IL, USA), R 4.1.1 (R Core Team).

## 3. Results

### 3.1. Cohort Characteristics

Of the 106 IBD patients and 42 healthy controls recruited, all subjects with suspected or confirmed SARS-CoV-2 infection were excluded from the study (*n* = 2 IBD patients, *n* = 3 healthy controls). All patients who were not vaccinated with mRNA vaccines (mRNA-1273/BNT162b2) or who were not inoculated with the same vaccine on both occasions were excluded (*n* = 11 IBD patients, *n* = 4 controls. Our post hoc power analysis showed a 1-beta error of 0.99. This value confirms the validity of this study.

IBD patients had a median age of 46 years and 53% were male. Most patients were vaccinated with the mRNA vaccines BNT162b2 (94%) or mRNA-1273 (6%). None of the patients and none of the controls died. There were no differences in body mass index (BMI) between IBD patients and healthy controls (*p* = 0.171). At the time points before the first/second vaccination and three months after the second vaccination, IBD patients showed normal values of inflammation (leukocytes, CRP, ferritin), renal retention (creatinine) and liver values (bilirubin, AST). Crohn’s disease was present in 63% of IBD patients (median CDAI score 0 [0–85]), and ulcerative colitis was present in 37% (median Mayo score 3 [0–5]). Oral mesalazine therapy was given in 46% of patients, and oral prednisolone therapy was used in 12%. Mesalazine therapy (supp.) was administered in 15% and budesonide therapy (supp.) was given in 10% of patients. Oral budesonide therapy was used in 4% of IBD patients. The most common comorbidity in the patients was pre-existing cardiovascular disease (22%), followed by renal failure (3%) (Table 1). According to the median, IBD patients had normal inflammatory parameters, renal retention parameters, and liver values (Appendix A). IBD patients were subdivided according to their current immunosuppressive therapies (*n* = 21 adalimumab, *n* = 31 infliximab, *n* = 15 vedolizumab, *n* = 18 ustekinumab, *n* = 9 others). Regarding patient characteristics (age, sex, and BMI), there were no significant differences in IBD patient subgroups. The proportional use of mRNA vaccines (BNT162b2, mRNA-12) was also the same in all subgroups (*p* = 0.273). There were no significant differences between the groups in terms of known pre-existing conditions and body mass index (BMI). Healthy controls and IBD patients were age-matched (Appendix A). In the group of healthy controls, fewer men were included (29% versus 53% females). Compared with IBD patients, healthy controls received the mRNA-1273 vaccine more frequently (6% IBD patients, 82% controls; *p* < 0.001) than the BNT162b2 vaccine (94% IBD patients, 18% controls; *p* < 0.001). None of the patients or controls had died before the end of the study (Appendix A).

### 3.2. Significantly Decreased SARS-CoV-2 S-IgG and sVNT Inhibition Levels in IBD Patients 3 Months after Second Vaccination

SARS-CoV-2 S-IgG (AU/mL) and sVNT results (% inhibition) were compared between the group of all IBD patients and healthy controls prior to the first and second vaccinations, as well as 3 and 6 months after the second vaccination. Before the first vaccination, all IBD patients and controls had no detectable antibodies (S-IgG, sVNT). Before the second vaccination and 3 months after the second vaccination, IBD patients showed significantly decreased antibody and receptor binding inhibition levels, respectively, compared with healthy controls (before second vaccination: sVNT 36% inhibition (19–63%) vs. 85% (64–89%), *p* < 0.001; S-IgG 329 AU/mL (86–1058 AU/mL) vs. 3285 AU/mL (1094–5385 AU/mL), *p* < 0.001; 3 months after second vaccination: sVNT 78% (38–95%) vs. 96% (95–97%), *p* = 0.002; S-IgG 1116 AU/mL (360–3214 AU/mL) vs. 4684 AU/mL (3552–10,630 AU/mL), *p* = 0.001) (Table 2). Six months after the second vaccination, IBD patients also showed decreased antibody levels compared with healthy controls, but these differences were barely not significant (sVNT 23% (15–95%) vs. 97% (85–97%), *p* = 0.062; S-IgG 104 AU/mL (0–3709 AU/mL) vs. 4289 AU/mL (1674–8730 AU/mL), *p* = 0.061). In the subgroup analysis of IBD patients with existing immunosuppressive therapies (adalimumab, infliximab, vedolizumab, azathioprine, ustekinumab, and others), there were no significant differences regarding S-IgG levels or sVNT values before the second vaccination and 3 months after the second vaccination (*p* > 0.05) (Figure 2a,b). At the timepoint prior to the second vaccination, significantly lower S-IgG levels were detected in IBD patients receiving immunosuppressive therapy with infliximab (161 AU/mL (34–435 AU/mL) vs. 3285 AU/mL (1094–5385 AU/mL), *p* = 0.028) and vedolizumab (649 AU/mL (51–1058 AU/mL) vs. 3285 AU/mL (1094–5385 AU/mL), *p* = 0.049) compared with healthy controls. The sVTN values of patients on immunosuppressive therapy with adalimumab (38% (8–68%) vs. 85% (64–89%), *p* = 0.002), infliximab (21% (16–37%) vs. 85% (64–89%), *p* < 0.001), and vedolizumab (47% (2–66%) vs. 85% (64–89%), *p* = 0.007) were significantly lower than those of healthy controls at this timepoint. Three months after the second vaccination, significantly lower sVNT inhibition levels were detected in IBD patients receiving anti-TNF therapy compared to healthy controls (adalimumab: 69% (27–90%) vs. 96% (95–97%), *p* = 0.023; infliximab: 73% (42–84%) vs. 96% (95–97%), *p* = 0.001; Figure 2c,d).

### 3.3. Seroconversion Rates in IBD Patients after SARS-CoV-2 Vaccination

Prior to the second COVID-19 vaccination, seroconversion as indicated by sVNT (inhibition > 30%) was detected in 54% of IBD patients (100% in healthy controls). Accordingly, the SARS-CoV-2 S-IgG levels (seropositivity > 49 AU/mL) before the second vaccination also indicated significantly higher conversion rates in the healthy controls than in the IBD patients (*p* < 0.001). Three months after the second vaccination, seroconversion was detected in 82% of IBD patients and 100% of healthy controls when looking at the sVNT conversion rate (*p* = 0.138). Furthermore, 98% of IBD patients showed serconversion regarding the SARS-CoV-2 S-IgG antibody level at this timepoint (100% controls, *p* = 0.847), demonstrating no significant different immune responses in either group. Six months after the second vaccination, the seroconversion rates of IBD patients were 40% as indicated by sVNT results (controls 100%, *p* = 0.045) and 60% with regard to SARS-CoV-2 S-IgG (controls 100%, *p* = 0.152), respectively (Figure 3a,b).

## 4. Discussion

Up to now, available studies revealed data about vaccination efficiency in IBD patients after the first vaccination and up to 12 weeks after the second vaccination, mainly considering S-IgG seroconversion. To the best of our knowledge, this study was the first one to investigate seroconversion rates and SARS-CoV-2 S-IgG and sVNT values in IBD patients 3 and 6 months after the second vaccination with mRNA vaccines, considering both S-IgG and sVNT conversion rates.

Including all IBD patients, irrespective of their respective immunosuppressive medication, seroconversion was detected in 53%/81% (sVNT/S-IgG) before the second vaccination. Patients on immunosuppressive therapy with adalimumab showed a sVNT seroconversion rate of 55% (S-IgG 82%), in contrast to those treated with infliximab with a rate of 35% (S-IgG 35%), vedolizumab with a rate of 57% (S-IgG 86%), and ustekinumab with a rate of 75% (S-IgG 92%), before the second vaccination. These results are consistent with the study by Kennedy et al. in which seroconversion after the first vaccination was detected in 20–27% of IBD patients on immunosuppressive therapy with infliximab and in 57–75% of patients on therapy with vedolizumab [21]. Three weeks after the initial vaccination, Reuken et al. were able to detect seroconversion in 71% of immunosuppressed IBD patients (steroids, TNF antibodies, vedolizumab, ustekinumab, azathioprine, mycophenolate, tacrolimus, and tofacitinib) [26]. However, our study showed significantly lower seroconversion rates before the second vaccination in IBD patients compared with healthy controls (sVNT *p* < 0.001, S-IgG *p* = 0.022). Reuken et al. were also able to demonstrate lower seroconversion rates in IBD patients at this timepoint compared with healthy controls, but this difference was not statistically significant [26]. Taken together, our results and the current literature confirm the urgent need for a second vaccination in IBD patients, as is already known for healthy persons. Furthermore, our results are in line with studies revealing attenuated vaccination efficiency in IBD patients on immunosuppressive therapies after vaccination against several other pathogens. In particular, immunomodulators, such as methotrexate and thiopurines, are known to attenuate serological responses against influenza vaccines, hepatitis B vaccines, and in case of pneumococcal vaccination [29,30,31]. However, additionally, anti-TNF agents, considered to be highly immunosuppressive, are able to reduce immunological responses, as was shown in studies investigating vaccination efficiency after a single vaccination against hepatitis A and B [32,33] in comparison to healthy controls. In contrast to anti TNF agents, IBD patients being treated with vedolizumab vaccine efficiency seem to not be affected after vaccination against influenza and hepatitis B [34,35], possibly due to its selective mechanism of action as antibody against the α4β7 integrin, which is mainly expressed in the gut [36].

Three months after the second vaccination, sVNT seroconversion was detected in 82% of the IBD patients, and S-IgG seroconversion was detected in 98%. There were no significant differences between patients on immunosuppressive therapies with adalimumab, infliximab, vedolizumab, and ustekinumab. A comparison with the healthy control group again showed no differences of the serum conversion rate in the immunosuppressed IBD patients (sVNT: seroconversion rate 82% IBD patients vs. 100% controls, *p* = 0.138; S-IgG: 98% IBD patients vs. 100% controls, *p* = 0.847). These data were in line with the results of Kappelmann et al. [20,23], who have demonstrated similar seroconversion rates in IBD patients and healthy controls. Taken together, our data confirm previous findings on the effectiveness of the second vaccination on humoral response in both IBD patients and healthy controls [22,23,24]. In addition, other studies on IBD patients highlighted the effectiveness of a second vaccination against other pathogens, such as influenza and hepatitis A [37,38], showing that vaccine effectiveness after the second vaccination even in patients on anti TNF agents is not attenuated compared to healthy controls.

Six months after the second vaccination, we detected a seroreactivity in sVNT in 40% of IBD patients and in S-IgG seroconversion in 60%. The rate of sVNT conversions in IBD patients was significantly lower than that of healthy controls (40% IBD patients vs. 100% controls; *p* = 0.045). Taking the low number of participants in the 6 month follow-up group into account as a limitation, our data indicate a significantly waning of humoral response in IBD patients compared with healthy controls. Further studies are necessary to investigate the efficacy of vaccination against SARS-CoV-2 after 6 months in IBD patients.

IBD patients receiving immunosuppressive therapy had significantly lower sVNT inhibition values than healthy controls at the timepoints before the second vaccination, as well as 3 and 6 months after the second vaccination. Looking at SARS-Co-2 S-IgG serum levels, IBD patients showed significantly lower levels at the timepoints before the second vaccination and 3 months after the second vaccination. S-IgG levels had also decreased in IBD patients 6 months after the second vaccination (*p* = 0.062). In subgroup analysis, IBD patients receiving infliximab and vedolizumab therapy had significantly lower S-IgG/sVNT values before the second vaccination compared with controls. Six months after the second vaccination, IBD patients on anti-TNF therapy (adalimumab, infliximab) also showed significantly lower sVNT levels. These results support the study by Kennedy et al. which demonstrated lower antibody levels in IBD patients receiving infliximab therapy 3–10 weeks after the first vaccination [21]. Wong et al. and Cerna et al. also demonstrated lower antibody levels in IBD patients on immunosuppressive therapy with anti-TNF therapy and vedolizumab [22,39]. Taken together, our study revealed a reduced humoral response in both S-IgG levels and sVNT inhibition levels in IBD patients, especially in patients treated with anti-TNF agents.

This study had some limitations. Of note, while serologic markers have been shown to correlate with vaccine success in terms of protective immunity, a quantitative threshold has not been defined so far [40]. This issue prompted us to characterize the potential neutralizing activity of patient sera by both quantitative anti-spike IgG assay and sVNT [27,28]. As stated by Perkmann et al., cutoff values given by manufacturers reflect diagnostic criteria of positivity; however, these do not indicate clear-cut protection from infection [40]. It must be mentioned that women were overrepresented in the healthy control group compared with the IBD patients (proportion of women in the IBD cohort versus the healthy control cohort was 71% versus 47%). Previous registration studies have shown comparable humoral responses following mRNA COVID-19 vaccination in males and females. In addition, a few pre-priming studies have reported sex differences in response to COVID-19 vaccination [41,42]. Some of the IBD patients had other pre-existing conditions (respiratory disease, kidney insufficiency, metastatic neoplasm, and diabetes) compared with the healthy controls. Due to the small number of patients in the “others” group, the results for this inhomogeneous subgroup are of limited value. The impact of preexisting diseases on vaccination efficiency against SARS-CoV-2 seems inconclusive. Although Naschitz et al. showed attenuated humoral responses in people with diabetes, cancer and with multiple morbidities [43], a recent study revealed comparable humoral responses in patients with type I and type II diabetes compared to healthy controls, but showed attenuated responses in case of renal insufficiency [44]. The IBD group was mostly vaccinated with BNT162b2, while the control cohort was predominantly vaccinated with mRNA-1273. Due to the prospective study design, which groups would be vaccinated with which vaccine was not predictable at the beginning of the study. Although employees at university hospitals were mostly vaccinated with mRNA-1273 at that time, vaccination centers and general practitioners predominantly used BNT162b2. Although Khan et al. found no significant differences between the two mRNA vaccines BNT16b2 and mRNA-1273 in IBD patients [45], Steensels et al. (2021) demonstrated higher antibody levels after mRNA-1273 vaccination compared with BNT162b2 in the period 6–10 weeks after the second vaccination when comparing the immune response in healthy volunteers [46]. Taken together, this might be a relevant confounder in the current study.

## 5. Conclusions

To our knowledge, this study is the first to demonstrate significantly lower SARS-CoV-2 S-IgG and sVNT values in IBD patients compared with healthy controls during a follow-up period of up to 6 months after the second vaccination, especially in patients on anti-TNF therapy. Although seroconversion rates did not differ significantly 3 months after the second vaccination, the current data indicate a relevant loss of humoral response in IBD patients 6 months after the second vaccination. The data highlight the importance of further prospective studies to evaluate humoral immunity in IBD patients 6 months and longer after the second vaccination and beyond; moreover, early booster shot vaccination should be discussed, especially for patients on anti-TNF therapy.

## Figures and Tables

**Figure 1 biomedicines-10-00171-f001:**
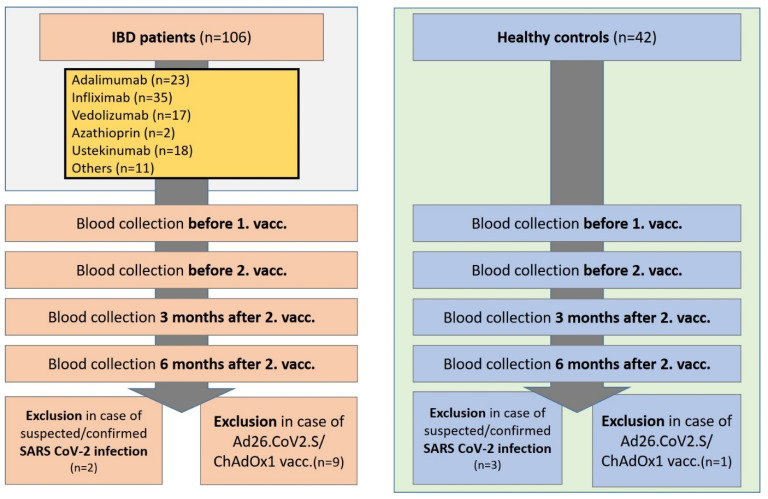
Study flow chart. Inclusion of IBD patients in the period from 01/2021 to 11/2021 on immunosuppressive medication and healthy controls. Blood collection within 48 h before the first and second vaccination, 3 and 6 months following the 2nd vaccination (+/− 7 days). Exclusion of patients and controls with suspected or confirmed SARS-CoV-2 infection, and ChAdOx1/Ad26.CoV2.S vaccination. IBD, inflammatory bowel disease; SARS-CoV-2, severe acute respiratory syndrome coronavirus type 2; vacc., vaccination.

**Figure 2 biomedicines-10-00171-f002:**
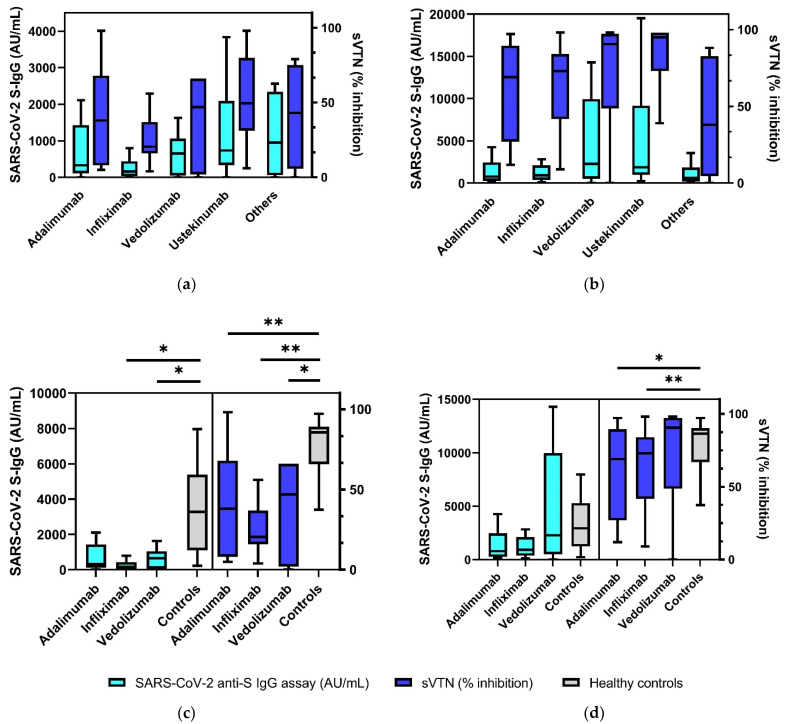
SARS-CoV-2-S IgG and sVTN inhibition levels in IBD patient subgroups before the second vaccination (**a**) and 3 months after the second vaccination (**b**). Comparison of antibody levels between IBD patients on immunosuppressive therapy (anti-TNF, vedolizumab) and healthy controls before the second vaccination (**c**) and 3 months after the second vaccination (**d**), antibody levels of all IBD patients in relation to existing immunosuppressive therapy compared with healthy controls (**e**). Tukey boxplots, * *p* > 0.05, ** *p* < 0.01.

**Figure 3 biomedicines-10-00171-f003:**
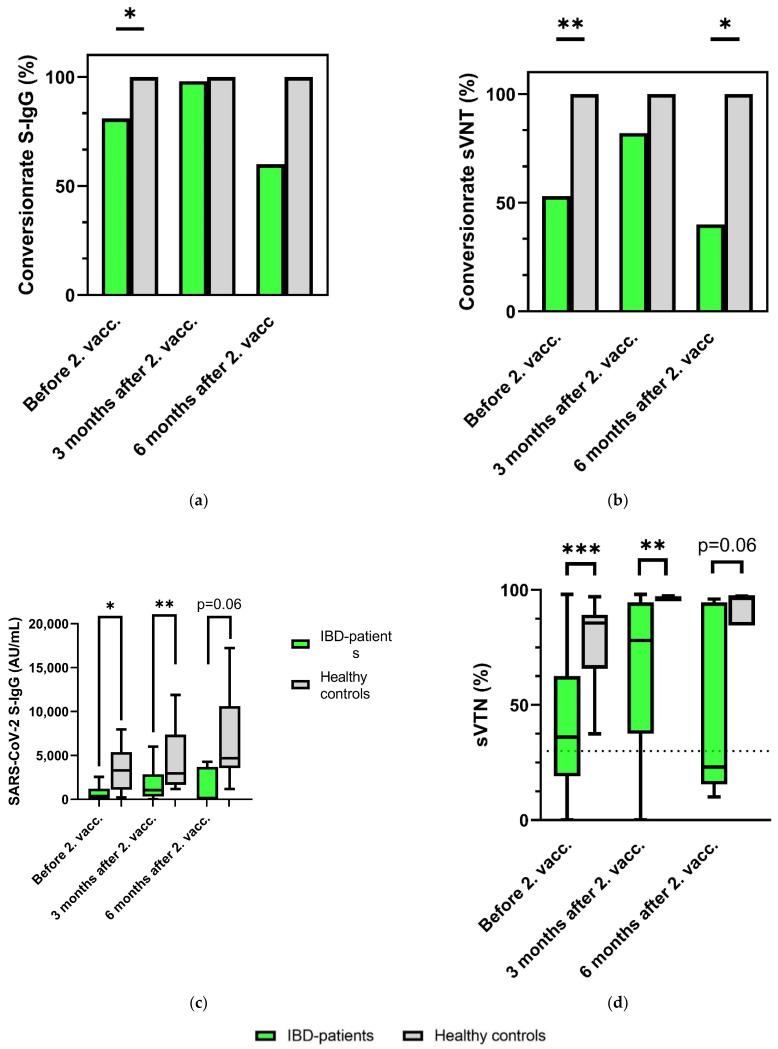
Humoral response of IBD patients and healthy controls before second vaccination, as well as 3 and 6 months after second vaccination. (**a**) Percentage seroconversion rates (sVTN > 30%); (**b**) percentage seroconversion rates (SARS-CoV-2 S-IgG ≥ 50 AU/mL); (**c**) quantitative detection of SARS-CoV-2 S-IgG; (**d**) percentage sVTN inhibition, representation of the cutoff values (30% inhibition) as a dashed line. IBD: inflammatory bowel disease; vacc: vaccination. Tukey boxplots, * *p <* 0.05, ** *p* < 0.01, and *** *p* < 0.001.

**Table 1 biomedicines-10-00171-t001:** Cohort characteristics of IBD patients. Representation of the entire IBD patient cohort and classification of patients by immunosuppressive therapy (adalimumab, infliximab, vedolizumab, azathioprine, ustekinumab, and others). The following immunosuppressants were summarized under “others”: corticosteroids, tofacitinib, certolizumab, golimumab, risankizumab, etrolizumab, mycophenolate mofetil, and azathioprine. IBD: inflammatory bowel disease; IQR: interquartile range; BMI: body mass index; SARS-CoV-2: severe acute respiratory syndrome coronavirus 2; CDAI score: Crohn’s disease activity index, p.o., per os.

Patients		IBD (*n* = 95)	Adalimumab (*n* = 21)	Infliximab (*n* = 31)	Vedolizumab (*n* = 15)	Ustekinumab (*n* = 18)	Others (*n* = 9)	*p*-Value
Patient characteristics	Age, years median (IQR)	46 (33–55)	37 (31–56)	49 (44–54)	46 (34–60)	41 (27–57)	33 (30–48)	0.093
	Sex, male (%)	50 (53)	13 (62)	15 (48)	10 (67)	7 (39)	4 (44)	0.474
	BMI	25 (22–28)	25 (22–28)	24 (22–27)	26 (21–31)	25 (22–29)	25 (23–28)	0.970
	Death (abs.)	0	0	0	0	0	0	1
SARS-CoV-2 vaccine	mRNA-1273 (%)	6 (6)	3 (14)	1 (3)	0 (0)	2 (11)	0 (0)	0.273
	BNT162b2 (%)	89 (94)	18 (86)	30 (97)	15 (100)	16 (89)	9 (100)	0.273
IBD	Crohn’s disease (%)	60 (63)	11 (52)	23 (74)	4 (27)	13 (72)	8 (89)	0.005
	CDAI score, median (IQR)	0 (0–85)	0 (0–103)	0 (0–10)	161 (79–233)	0 (0–130)	0 (0–148)	0.025
	Ulcerative colitis (%)	35 (37)	10 (48)	8 (26)	11 (73)	5 (28)	1 (11)	0.005
	Mayo score, median (IQR)	3 (0–5)	0 (0–3)	3 (1–4)	4 (2–7)	5 (3–9)	0 (0–0)	0.044
Medication	Prednisolone p.o. (%)	11 (12)	0 (0)	1 (3)	4 (27)	3 (17)	3 (33)	0.018
	Budesonide p.o. (%)	4 (4)	0 (0)	2 (7)	0 (0)	1 (6)	1 (11)	0.578
	Budesonide supp. (%)	9 (10)	0 (0)	2 (7)	5 (33)	1 (6)	1 (11)	0.012
	Mesalazine p.o. (%)	44 (46)	11 (52)	11 (36)	9 (60)	7 (39)	6 (67)	0.399
	Mesalazine supp (%)	14 (15)	3 (14)	1 (3)	4 (27)	4 (22)	2 (22)	0.200
Pre-existing conditions	Cardiovascular disease	21 (22)	3 (14)	9 (29)	6 (40)	2 (11)	1 (11)	0.164
	Respiratory disease (%)	9 (10)	1 (5)	2 (7)	4 (27)	1 (6)	1 (1)	0.173
	Kidney insufficiency (%)	3 (3)	0 (0)	2 (7)	1 (7)	0 (0)	0 (0)	0.509
	Metastatic neoplasm (%)	1 (1)	0 (0)	1 (3)	0 (0)	0 (0)	0 (0)	0.720
	Diabetes (%)	3 (3)	0 (0)	2 (7)	1 (7)	0 (0)	0 (0)	0.509
	Hematologic malignancy (%)	0 (0)	0 (0)	0 (0)	0 (0)	0 (0)	0 (0)	1

**Table 2 biomedicines-10-00171-t002:** SARS-CoV-2 S-IgG levels and sVNT values of IBD patients and healthy controls before the first and second vaccination, as well as 3 and 6 months after the second vaccination. Partitioning of the IBD patient population was completed based on immunosuppressive therapy (adalimumab, infliximab, vedolizumab, and ustekinumab). Other immunosuppressive therapies (corticosteroids, tofacitinib, certolizumab, golimumab, risankizumab, etrolizumab, mycophenolate mofetil, and azathioprine) were grouped under “others”. IBD: inflammatory bowel disease; IQR: interquartile range; vacc.: vaccination.

Patients		IBD (*n* = 95)	Controls (*n* = 38)	*p*-Value	Adalimumab (*n* = 21)	Infliximab (*n* = 31)	Vedolizumab (*n* = 15)	Ustekinumab (*n* = 18)	Others (*n* = 9)	*p*-Value
Before 2nd vaccination	Samples (n)	52	22		11	17	7	12	4	
	SARS-CoV-2 S-IgG (AU/mL), median (IQR)	329 (86–1058)	3285 (1094–5385)	<0.001	329 (110–1428)	161 (34–435)	649 (51–1058)	732 (334–2091)	952 (58–2338)	0.136
	Seroconversion rateS-IgG (%)	81	100	0.022	82	72	86	92	75	0.742
	sVNT (%), median (IQR)	36 (19–63)	85 (64–89)	<0.001	38 (8–68)	21 (16–37)	47 (2–66)	50 (31–80)	43 (6–75)	0.201
	Seroconversion rate sVNT (%)	53	100	<0.001	55	35	57	75	50	0.337
3 months after 2nd vaccination	Samples (n)	60	11		13	20	10	11	6	
	SARS-CoV-2 S-IgG (AU/mL), median (IQR)	1116 (360–3214)	4684 (3552–10,630)	0.001	777 (258–2451)	920 (367–2100)	2281 (510–9964)	1867 (988–9177)	609 (198–1841)	0.176
	Seroconversion rateS-IgG (%)	98	100	0.847	100	100	90	100	67	0.394
	sVNT (%), median (IQR)	78 (38–95)	96 (95–97)	0.002	69 (27–90)	73 (42–84)	90 (48–97)	95 (73–98)	38 (5–83)	0.093
	Seroconversion rate sVNT (%)	82	100	0.138	69	85	90	91	67	0.585
6 months after 2nd vaccination	Samples (n)	4	7		1	3	0	0	0	
	SARS-CoV-2 S-IgG (AU/mL), median (IQR)	104 (0–3709)	4289 (1674–8730)	0.061	104	0 (0–0)				0.755
	Seroconversion rate S-IgG (%)	60	100	0.152	100	33				0.329
	sVNT (%), median (IQR)	23 (15–95)	97 (85–97)	0.062	21	23 (10–23)				0.344
	Seroconversion rate sVNT (%)	40	100	0.045	0	33				0.392

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
