# Peer review of "Humoral Immune Response in IBD Patients Three and Six Months after Vaccination with the SARS-CoV-2 mRNA Vaccines mRNA-1273 and BNT162b2"

_biomedicines, 2022, doi:10.3390/biomedicines10010171_

Round 1
Reviewer 1 Report
In the present original article Vollenberg et al shake that, in a group of 95 inflammatory bowel disease (IBD) patients, the humoral response to mRNA-based vaccines against SARS-CoV2 is weaker than in healthy controls and, in particular, those treated with anti-TNF agents had significantly lower antibody levels after the second dose. Main comments:
1) Lines 93-94: this is a result, not a method
2) Line 102: cortisone - - > corticosteroids
3) A linguistic revision is necessary (see for example “applied” line 165 or “renal insufficiency” line 169).
4) Were patients with previous SARS-CoV2 infection undergoing vaccine included? Please explain in the inclusion/exclusion criteria section.
5) Sample size calculation is absent.
6) The statistical tests reported in paragraph 2.4 are based on the assumption that data distribution is not Gaussian. Was Kolmogorov Smirnov test performed to prove that?
7) Some disease data reported in supplementary tables S2-S3 are very important and should be integrated in the main text.
8) Few patients were on conventional immunosuppressive therapy (azathioprine). This could be a bias.
Reviewer 2 Report
The present study investigates the humoral response in immunosuppressed IBD patients after mRNA COVID vaccination. The results presented in the manuscript indicate a relevant loss of humoral response in IBD patients treated with anti-TNFa therapy 6 months after the second vaccination. Overall, the manuscript is added important information to the field of research. However, as a reviewer, I have a few points here that should be considered carefully.
- The abstract should follow the Biomedicines guideline to authors. The abstract should be a total of about 200 words and should follow the style of structured abstracts, but without headings: Background, Methods, Results and Conclusion.
- In the Methods section, I suggest adding the study design scheme to help understand the procedures and timeline of the investigation.
- In table 1, the authors present the experimental group characteristic. The characteristic should also include a control group. Please complete subsection 3.1 and table 1 with the characteristics of the control group.
- Additionally, in the experimental group characteristic, it is a lack of information about BMI in control and IBD patients. Obesity also influences the immune response after vaccination.
- Figure 1 should appear just after the first citation in the text. Additionally, why did the authors place Figure 2 before Figure 1. The references to the figures and figures number should be double-checked.
- A few times, the authors describe the results statistically non-significant (e.g. lines 201-204). If the statistical analysis revealed no statistically significant changes, the observed alterations are not different. The result 3.2 section should be rewritten to describe only significant results.
- While the serum experiments seem well designed, the presentation of the results would benefit from another organization. Please, consider presenting time points for each treatment on the X-axis in figure 1. Additionally, Figures 1 a and b did not include the results of the control group (authors referenced this figure when the results were described compere with healthy controls). Please consider adding the control group to the Figure 1 and b.
- What do the error bars mean in Figure 1 and 2. Whether it is SD or SE? It should be described in the figure caption. The error bars should be added in Figure 2 (a, b).
- In the first sentence of the discussion the authors pointed out that their study for the first time investigated seroconversion in IBD patients while a few lines later (258-263) the authors have written that the obtained results are consistent with other studies. Please indicate the novelty of the present study in the light of other investigations.
- The discussion is largely a repetition of the description of the results. The authors should precede it with a physiological explanation of the studied processes and discuss their own results with the results of other authors (as it is).
- Similarly, the authors described the results statistically nonsignificant in the discussion results sections. E.g. line 273-277 “A comparison with the healthy control group again showed a lower serum conversion rate in (…) but these results were not statistically significant”. It should be corrected. P-value 0,847 is even not close to significant.
- The authors should discuss the proposed mechanism responsible for lower humoral response in patients with IBD, passed on IBD pathophysiology, and in patients on immunosuppressive therapy.
- Additionally, authors should discuss how pre-existing conditions, including diabetes, may influence the results.
Round 2
Reviewer 1 Report
Answers were satisfactory
Reviewer 2 Report
All my suggestions were met, thank you.